# SE-VLN: A Self-Evolving Vision-Language Navigation Framework Based on Multimodal Large Language Models

## Abstract

Recent advances in vision-language navigation (VLN) were mainly attributed to emerging large language models (LLMs). These methods exhibited excellent generalization capabilities in instruction understanding and task reasoning. However, they were constrained by the fixed knowledge bases and reasoning abilities of LLMs, preventing fully incorporating experiential knowledge and thus resulting in a lack of efficient evolutionary capacity. To address this, we drew inspiration from the evolution capabilities of natural agents, and proposed a self-evolving VLN framework (SE-VLN) to endow VLN agents with the ability to continuously evolve during testing. To the best of our knowledge, it was the first time that a multimodal LLM-powered training-free self-evolving VLN framework was proposed. Specifically, SE-VLN comprised three core modules, i.e., a hierarchical memory module to transfer successful and failure cases into reusable knowledge, a retrieval-augmented thought-based reasoning module to retrieve experience and enable multi-step decision-making, and a reflection module to realize continual evolution. Comprehensive tests illustrated that the SE-VLN achieved navigation success rates of 57% and 35.2% in unseen environments, representing relative performance improvements of 23.9% and 15.0% over current state-of-the-art methods on R2R and REVERIE datasets, respectively. Moreover, the SE-VLN showed performance improvement with increasing experience repository, elucidating its great potential as a self-evolving agent framework for VLN. Project page: `https://anonymous.4open.science/r/SE-VLN-76B2/README.md`

## 1 Introduction

Vision-language navigation (VLN), as a key technology connecting human natural language and robot visual navigation, aims to enable embodied agents to autonomously plan paths and complete visual navigation tasks in unseen environments based on human language instructions Anderson et al. (2018). Despite its rapid development Chen et al. (2024b); Schumann et al. (2024); Li et al. (2024b); Lin et al. (2024), especially with the help of emerging LLMs Zhou et al. (2024); Chen et al. (2024a); Li et al. (2024a); Long et al. (2024), current VLN methods lack the advanced navigation capabilities of natural agents (e.g., bats, migratory birds, humpback whales) and shows critical shortcomings in autonomous evolution through experience. In fact, an experienced horse refines its neural circuitry through repeated journeys. Its navigation skill distills wisdom from experience and evolves autonomously through adaptation, providing a biological blueprint for existing VLN to overcome bottlenecks of "data dependency" and "scene generalization".

Current VLN methods Li et al. (2024a); Zhou et al. (2024); Chen et al. (2024a) had significant limitations in terms of static experience utilization. They treated historical trajectory data as a fixed replay buffer, only using it to maintain decision consistency for the current task, and failed to extract reusable general knowledge from dynamic processes. Another drawback of existing VLN methods Chen et al. (2024a); Zhou et al. (2024) lay in their limited reasoning capabilities. They relied on the static knowledge invocation of pre-trained models and failed to effectively integrate experience for multi-step decision-making. This stood in stark contrast to the biological intelligence of natural agents such as horses, which adjusted navigation paths by combining environmental memories with

real-time perceptions, helping gain precise alignment between visual perceptions and language instructions when processing complex commands. Furthermore, the "evolution" processes of existing VLN agents were highly dependent on manual hyperparameter tuning Anderson et al. (2018); Lin et al. (2024) and model iteration Li et al. (2024c); Lin et al. (2025), which was completely different from the mechanism which biological agents used to achieve self-evolution through natural selection and neural plasticity.

To bridge this gap, this paper proposed a training-free VLN framework, which integrated three core modules, i.e., a hierarchical memory module, a retrieval-augmented thought-based reasoning module, and a reflection module, to realize a multimodal LLM-powered self-evolving VLN framework. This paper designed a hierarchical memory module to store short-term memory and long-term experience. It constructed a verbal topological map to record the agent's visual observations and decision-making processes at each node, providing foundational support for real-time decision-making and experience extraction. Meanwhile, it used an experience repository to store long-term accumulated navigation experience, thereby enhancing the decision-making efficiency of subsequent tasks. This paper also introduced a retrieval-enhanced thought-based reasoning module. This module utilized retrieval-augmented generation (RAG) technology to retrieve historical experience related to the current task from the experience repository and combined chain-of-thought (CoT) prompting to decompose multi-source information into executable multi-step reasoning chains, wherein verbal topological map was dynamically updated and decisions were better made. Further, We introduced a reflection module, which conducted in-depth analysis of the agent's decisions guided by task evaluation results, enabling dynamic updating and continual evolution.

Extensive experiments on R2R and REVERIE datasets demonstrated that the SE-VLN achieved state-of-the-art performance and exhibited self-evolving VLN ability as the experience repository expanded.

Our work mainly contributed to the following:

- A training-free self-evolving VLN framework was proposed by simulating the evolution processes of natural agent navigation with the help of MLLM.
- A hierarchical memory module was proposed to record the agent's visual observations and decision-making processes at each path node through a verbal topological map, supporting immediate decision-making and experience refinement. It was also combined with an experience repository to store long-term experience, enhancing decision-making for subsequent tasks.
- A retrieval-augmented thought-based reasoning module integrated with retrieval-augmented generation (RAG) and chain-of-thought (CoT) was introduced to enable multi-step decision-making by retrieving relevant historical experience, thereby enhancing the accuracy of the agent's decisions.
- A reflection module was introduced to conduct in-depth analysis of the agent's decisions based on task evaluation results, enabling incremental updates of long-term experience and promoting the agent's continuous evolution.

## 2 RELATED WORK

### 2.1 VISION-LANGUAGE NAVIGATION (VLN)

Researchers had extensively explored VLN methods that integrated memory and reasoning mechanisms to enhance the efficiency and robustness of agent navigation Zhou et al. (2024); Chen et al. (2024b); Schumann et al. (2024); Li et al. (2024b). The memory module played a crucial role in VLN agents, enabling them to store and review historical information such as explored paths Chen et al. (2024a), identified landmarks Zhan et al. (2024), or past observation data Li et al. (2024a). This capability was vital for handling long-range navigation tasks and making informed decisions in non-Markovian environments, as agents could utilize memory to optimize path selection and more effectively understand the connections between the current environment and historical states Chen et al. (2024a); Zhan et al. (2024). Meanwhile, the reasoning module endowed agents with the ability to deeply understand language instructions and analyze environmental visual information. Agents could decompose abstract natural language instructions into a series of executable sub-goals and

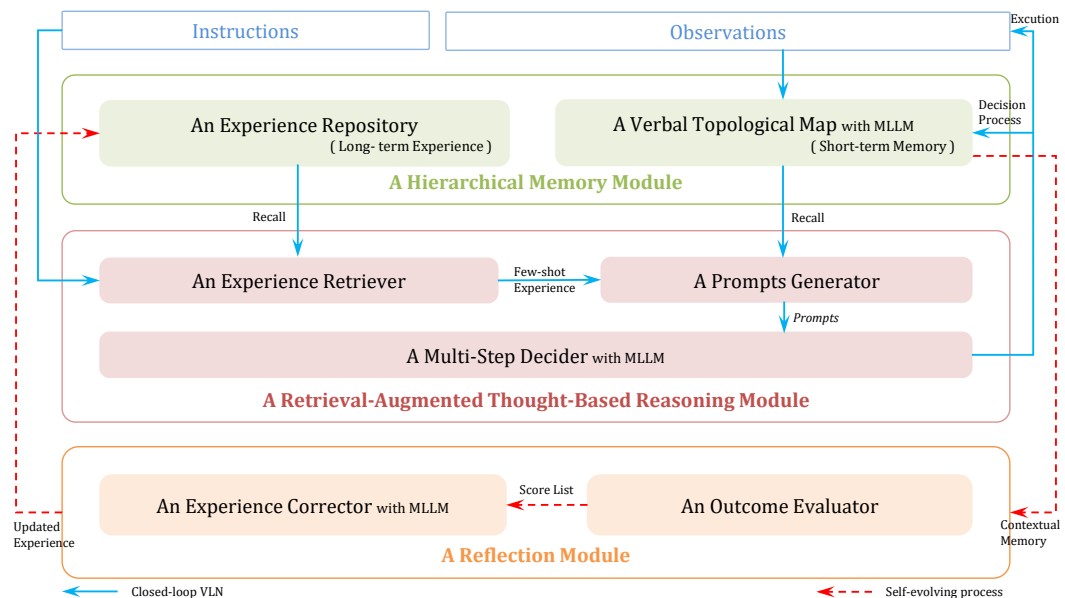

Figure 1: The workflow of SE-VLN.

perform semantic analysis on visual inputs to identify relevant objects, comprehend spatial relationships, and infer feasible paths Chen et al. (2024a); Zhan et al. (2024). This strong reasoning capability allowed agents to make rational decisions even when faced with ambiguous or incomplete instructions, and to extract key clues from visual scenes that guided navigation. However, these methods mostly relied on the inherent knowledge bases and reasoning capabilities of LLMs to improve navigation performance.

### 2.2 LARGE LANGUAGE MODELS (LLMs) FOR AGENTS

Recent research leveraged LLMs and VLMs as agents, particularly in the domains of gaming, robotics, and healthcare, not only to provide rigorous evaluation platforms for state-of-the-art AI systems but also to herald transformative impacts of agent-centric AI on society and industries. For instance, Lan et al. Lan et al. (2023) utilized system prompts to guide LLMs in gaming, achieving multi-agent collaboration and competition in Avalon. Han et al. Han et al. (2024) developed an LLM-based task planner to enhance human preference alignment in home service robots to better understand and adapt to individual users' specific needs and preferences. However, these agents generally lacked the capability for self-evolution and continual learning. To address this limitation, Shinn et al. Shinn et al. (2024) introduced a mechanism that involved verbal reflection on task feedback signals and recorded these reflections in an episodic memory buffer to guide more optimal decision in subsequent trials. Zhang et al. Zhang et al. (2024) devised a strategy-level reflection and optimization method that iteratively reviewed past actions and employed depth-first search techniques for policy optimization, thereby facilitating the continuous evolution of agents in poker games. Despite significant progress in their respective tasks, constructing LLM-powered VLN agents with continuous evolution remains an unexplored area.

## 3 METHODOLOGY

### 3.1 PRELIMINARIES

In a discrete environment with an undirected navigation graph $\mathcal{G}$, the agent needed to navigate from a starting node to a target node according to natural language instruction $\mathcal{I}$, which consisted of a sequence of words $\{i_1, i_2, i_3, \ldots, i_n\}$ with potential landmarks along the way and action commands to be executed. At step $t$, the agent received visual observations $\mathcal{O}_t = \{\mathcal{O}_{t,k}\}_{k=1}^{K}$ from the simulator, where $K$ denoted that $\mathcal{O}_t$ contained $K$ navigable viewpoints. The agent predicted the next action,

which was then executed through interaction with the simulator to transition to the next location. The task was considered successful when the agent moved within a 3-meter range of the target location.

## 3.2 OVERALL ARCHITECTURE

Initially, the agent used an experience retriever based on the instructions of the current task to fetch few-shot experience from the experience repository, used for decision-making at each step, as shown in Figure 1. Then, environmental observations were updated into a verbal topological map, where a prompt generator integrated information and constructed prompts. After comprehensive analysis through a multi-step decider, actions were output to interact with the environment, while the decision process was simultaneously updated into the verbal topological map. This process was executed in a closed loop during the task. Meanwhile, an outcome evaluator of the reflection module used the contextual memory of the verbal topological map to quantitatively assess navigation performance. Subsequently, an experience corrector identified and corrected unreasonable decisions by combining score lists, saved the corrected decisions as experience back into the experience repository, and achieved self-evolution by learning from historical experience.

## 3.3 A HIERARCHICAL MEMORY MODULE

The hierarchical memory module, comprising a verbal topological map and an experience repository, enabled the agent to retrieve contextual memory for current task and past similar experience, which were crucial for enhancing the navigation performance.

### 3.3.1 A VERBAL TOPOLOGICAL MAP

The verbal topological map served as short-term memory for enhancing navigation performance. This map recorded key information in real-time, including navigation graph $\mathcal{G}_t \subset \mathcal{G}$, textual descriptions $\mathcal{D} = \{\mathcal{D}_1, ..., \mathcal{D}_t\}$ of visual observations $\mathcal{O} = \{\mathcal{O}_1, ..., \mathcal{O}_t\}$, and decision processes $\langle \mathcal{T}, \mathcal{P}, \mathcal{A} \rangle$, which included thinking $\mathcal{T} = \{\mathcal{T}_1, ..., \mathcal{T}_t\}$, planning $\mathcal{P} = \{\mathcal{P}_1, ..., \mathcal{P}_t\}$ and executing $\mathcal{A} = \{\mathcal{A}_1, ..., \mathcal{A}_t\}$. This also enabled it to serve as the basis for analyzing erroneous decisions within the reflection module. Details can be found in Figure. 6 in Appendix.

Its construction involved following two key processes.
**Topological Mapping:** In the VLN task, the agent had never explored the entire environment and must construct a map based on its online observations. Therefore, we stored the map as a dynamically updated graph. At step $t$, we recorded all nodes visited by the agent and their connectivity in the graph $\mathcal{G}_t$. The agent selected the next node to visit from several currently navigable viewpoints provided by the simulator and updated the graph from $\mathcal{G}_t$ to $\mathcal{G}_{t+1}$.
**Map Annotations:** To assist the agent in navigating more effectively, we added annotations to each node of the topological graph $\mathcal{G}_t$, creating contextual memory. This enabled the agent to refer to them when making decisions. At step $t$, we converted the agent's visual observations $\mathcal{O}_t$ at the current topological node $V_t$ into textual descriptions $\mathcal{D}_t$, which became part of the node annotations. After the reasoning module made a decision, we also saved the decision process $\langle \mathcal{T}_t, \mathcal{P}_t, \mathcal{A}_t \rangle$ as part of the map annotations. These map annotations aided in decision at the next step and post-task reflection. Consequently, the map $\mathcal{M}_t$ could be represented using the following paradigm:

$$\mathcal{D}_t = MLLM\left(\mathcal{S}_M, \mathcal{O}_t\right) \tag{1}$$

$$\mathcal{M}_t = \underbrace{\mathcal{G}_t}_{\substack{\text{topological} \\ \text{graph}}} \oplus \underbrace{\mathcal{D}}_{\substack{\text{scene} \\ \text{descriptions}}} \oplus \underbrace{\mathcal{T} \oplus \mathcal{P} \oplus \mathcal{A}}_{\substack{\text{Posterior decision} \\ \text{annotation}}} \tag{2}$$

where $\mathcal{S}_M$ represented the task descriptions for the hierarchical memory module.

### 3.3.2 AN EXPERIENCE REPOSITORY

The experience repository $\mathcal{E}_{DB}$, built on the vector database Chroma, documented the agent's navigation memory from past tasks as long-term experience to guide subsequent task decisions. Each experience $e$ comprised four key elements: landmark features $\mathcal{L}$, scene descriptions $\mathcal{D}$, decision processes $\langle \mathcal{T}, \mathcal{P}, \mathcal{A} \rangle$, and revised decision processes $\langle \mathcal{T}', \mathcal{P}', \mathcal{A}' \rangle$. Each experience $e$ entry could be

formally defined as a quadruple:

$$e = \langle \mathcal{L}, \mathcal{D}, \langle \mathcal{T}, \mathcal{P}, \mathcal{A} \rangle, \langle \mathcal{T}', \mathcal{P}', \mathcal{A}' \rangle \rangle \tag{3}$$

Landmark features $\mathcal{L}$ referred to the destinations included in the instruction. The agent used them to retrieve similar experience for reasoning. Scene descriptions $\mathcal{D}$ recorded textual depictions of the agent's visual observations at each decision frame, aiding the agent in better understanding and responding to complex navigation environments. The decision processes $\langle \mathcal{T}, \mathcal{P}, \mathcal{A} \rangle$ logged the agent's thinking, planning, and executing at each decision frame. The agent could learn from past experience, thereby making more accurate and effective judgments.

### 3.4 A RETRIEVAL-AUGMENTED THOUGHT-BASED REASONING MODULE

The retrieval-augmented thought-based reasoning module served as the decision-making core of SE-VLN. It incorporated retrieval-augmented generation (RAG) technology to retrieve experience, generate prompts, and combine chain-of-thought (CoT) prompting technology for multi-step decision-making. Specifically, it involved an experience retriever performing few-shot experience retrieval, a prompt generator integrating historical information to generate prompts, and a multi-step decider executing interpretable reasoning and interacting with the environment. Details can be found in Figure 7 in Appendix.

#### 3.4.1 AN EXPERIENCE RETRIEVER

As the number of navigations increased, the experience repository would accumulate a large amount of navigation experience. Inputting all experience as prompts would lead to a dimensionality explosion problem:

$$\lim_{N \to \infty} \mathcal{H}(P_{\text{full}}) \propto O(N \cdot d) \tag{4}$$

Where $\mathcal{H}$ represented the entropy of the prompt information, and $d$ was the embedding dimension of a single experience.

To reduce the risk of decision-making confusion in large language models, we constructed an experience retriever based on semantic similarity. This retriever extracted historical experience with the same landmark features from the experience repository based on the core landmark features contained in the instruction of the current task. This design was primarily based on two considerations: 1) complete instruction usually contained multiple semantic elements such as destinations, items, actions, etc., with the presence of noise interference factors, and 2) the perception features and decision sequences were highly reusable under the same landmark scenario. Details can be found in Appendix A.1.

#### 3.4.2 A PROMPT GENERATOR

To help the VLN agent better understand tasks and make decisions, we developed a prompt generator that organized various pieces of information into four key components for each decision frame: task descriptions $\mathcal{S}_R$, which provided the background of the VLN task along with definitions of inputs and outputs, including their formats and constraints on the reasoning process; instruction $\mathcal{I}$ that specified the actions to be performed and the destination to reach; the contextual memory of a verbal topological map $\mathcal{M}_t$ that recorded the agent's short-term memory, aiding in processing real-time contextual information for short-term reasoning and decision-making; and few-shot experience $\mathcal{E}$ containing similar cases involving comparable sensory data and decision paths, allowing the agent to learn and refer to them for more informed decisions. In summary, at each decision step, we built a tailored prompt based on the current memory to guide the decision-making process. It could be formalized as:

$$PG_t = \Phi\left(\mathcal{S}_R, \mathcal{I}, \Psi(\mathcal{M}_t), \mathcal{E}\right) \tag{5}$$

Where $PG_t$ was the prompt at step $t$, $\Phi$ was the prompt generation operator, responsible for merging information from multiple sources and structuring it to generate the final prompt that guided the decision-making of the VLN agent. $\Psi$ was the dynamic memory encoding function used to transform the raw memory data at step $t$ into a structured representation for prompt generation.

### 3.4.3 A MULTI-STEP DECIDER

The multi-step decider based on chain-of-thought (CoT) reasoning was an essential component, with its core mechanism embodied in breaking down complex decisions into explainable and progressive reasoning steps through CoT. On one hand, it could illustrate the logic and process of reasoning, enhancing the transparency and accuracy of decisions. On the other hand, it could serve as part of short-term memory, used for error correction by the reflection module and for enhancing the consistency of real-time decision-making. The entire process was divided into three stages: thinking, planning, and executing.

**Thinking** $\mathcal{T}$: In this stage, the agent synthesized and analyzed input information by generating detailed reasoning steps to determine the next action to take and the rationale behind it, providing a solid foundation for subsequent planning.

**Planning** $\mathcal{P}$: Based on the analysis results from the thinking stage, the agent developed specific paths and action plans for subsequent decision-making.

**Executing** $\mathcal{A}$: In this stage, the agent executed concrete actions according to the path generated during the planning stage, interacting with the environment.

To formalize this process, we could use the following equation to represent the reasoning at step $t$:

$$\langle \mathcal{T}_t, \mathcal{P}_t, \mathcal{A}_t \rangle = MLLM(PG_t) \tag{6}$$

### 3.5 A REFLECTION MODULE

In order to accumulate valuable experience and enrich the experience repository after each navigation task, we designed a reflection module akin to how humans learn and grow to correct unreasonable decisions made during the navigation. However, we found that directly requiring the MLLM to reflect on the agent's lengthy navigation process would cause the agent to lose focus on key issues, thereby affecting the effectiveness and accuracy of the reflection. Therefore, we divided the reflection module into two components. Details can be found in Figure 8 in Appendix.

#### 3.5.1 AN OUTCOME EVALUATOR

In VLN tasks, the most commonly used metrics for evaluating an agent's navigation performance are navigation error (NE), success rate (SR), path length weighted success rate (SPL), and oracle success rate (OSR). These metrics provide a direct reflection of the agent's capability to reach the target location and its navigational efficiency. Adhering to this established framework, we utilized ground truth data from the MatterPort3D simulator to accurately compute the outcomes of the current navigation tasks. This approach not only ensured the precision of the results but also facilitated the reflection module within the agent, enabling it to specifically identify and correct unreasonable decisions made during navigation. In the real world, we could evaluate the navigation performance of the agent through interactive feedback with human experts.

$$\mathcal{E}_{\text{metric}} = [\text{NE, SR, SPL, OSR}] = f_{\text{eval}}(\tau_{\text{nav}}, \tau_{\text{gt}}) \tag{7}$$

Where $\mathcal{E}_{\text{metric}}$ denoted the value of the evaluation metrics, $f_{\text{eval}}$ represented the evaluation function, $\tau_{\text{nav}}$ referred to the navigation trajectory of the agent, and $\tau_{\text{gt}}$ indicated the ground truth path provided by the simulator.

#### 3.5.2 AN EXPERIENCE CORRECTOR

In order to enable the VLN agent to learn from its experience autonomously, we used MLLM as error correctors. Specifically, we utilized the contextual memory from the verbal topological map as a prompt for the MLLM, instructed the MLLM to analyze the evaluation results, identified the reasons for the first unreasonable decision, and provided the correct decision process. Finally, we embedded the landmark features as a key, paired them with contextual memory and correct decision process to form experience, and stored these in the experience repository.

$$\langle \mathcal{T}', \mathcal{P}', \mathcal{A}' \rangle = MLLM\left(\mathcal{S}_{\text{ref}}, \Psi(\mathcal{M}_t), \mathcal{E}_{\text{metric}}\right) \tag{8}$$

$$\mathcal{E}_{\text{DB}} \leftarrow \mathcal{E}_{\text{DB}} \cup e_{\text{new}} \tag{9}$$

Where $\mathcal{S}_{\text{ref}}$ referred to the task descriptions of the reflection module, $\mathcal{E}_{\text{DB}}$ referred to the experience repository , and $e_{\text{new}}$ referred to the new experience.

# 4 EXPERIMENTS

## 4.1 DATASETS

The R2R Anderson et al. (2018) and REVERIE Qi et al. (2020) datasets are widely used for evaluating system performance in VLN tasks. The R2R dataset was built upon 90 real indoor environments within the Matterport3D simulator, encompassing residences, apartments, and office spaces. It comprised 7,189 trajectories, with each trajectory associated with three fine-grained instructions, resulting in a total of 21,567 navigation instructions, averaging 29 words per instruction. These instructions aimed to guide agents through complex layout understanding and multi-step decision processes from one room to another. In contrast to the R2R dataset, which focused on fine-grained room-to-room navigation instructions, the REVERIE dataset posed a higher challenge by introducing the task of locating specific target objects, thereby demanding enhanced environmental understanding and reasoning skills from the agent. Similarly constructed within 90 real indoor environments in the Matterport3D simulator, REVERIE included 4,140 target objects and 21,702 crowd-sourced instructions, averaging 18 words per instruction. This dataset required agents to receive natural language instructions at a starting position, directing them to a remote target object located elsewhere within the same building.

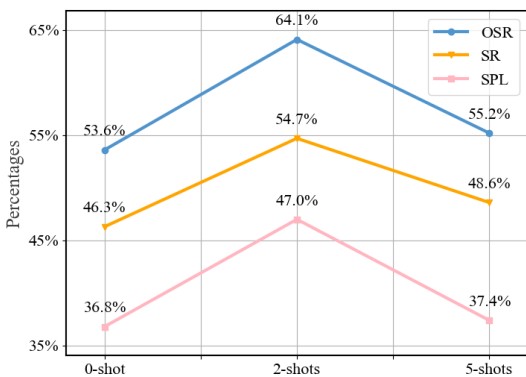

Figure 2: The impact of different shots experience.

## 4.2 EVALUATION METRICS

The following standard metrics were used to evaluate the performance of VLN tasks on the R2R and REVERIE datasets:

*1) Oracle Success Rate (OSR)*: This represented the proportion of times the agent could successfully complete the task when provided with the optimal path or guidance.

*2) Success Rate (SR)*: This referred to the proportion of times the agent successfully completes the navigation task. Specifically, a navigation attempt was considered successful if the agent stopped within 3 meters of the target point.

*3) Success weighted by Path Length (SPL)*: This metric evaluated the efficiency of the agent's navigation by comparing the length of the path taken by the agent to the optimal path.

*4) Navigation Error (NE)*: This measured the distance between the agent's final position and the target location, with shorter distances indicating more accurate navigation.

## 4.3 IMPLEMENTATION DETAILS

We conducted experiments based on current mainstream MLLMs, including Claude 3.5 Sonnet, Gemini-2.0-Pro, Qwen2-VL, InternVL2.5, and GPT-4o, to verify the applicability and effectiveness of different models within this framework. Among them, GPT-4o demonstrated the best performance under the SE-VLN framework, so we selected GPT-4o (with a contextual window length of 128K tokens) to present the experimental results. Details can be found in Figure 5 in Appendix.

## 4.4 ABLATION STUDY

We conducted extensive ablation studies on a subsample of R2R, comprising 72 scenes from the R2R dataset with 216 trajectories, to analyze the impact of each key module in the SE-VLN.

### 4.4.1 EFFECT OF LONG-TERM EXPERIENCE

As shown in Figure 2, we evaluated the impact of numbers of experience inputs on navigation performance by incorporating them into the reasoning module. In this study, "0-shot" denoted the scenario where no past experience were utilized; "2-shots" and "5-shots" referred to introducing two and five of the most similar past experience, respectively, as prompts before inference for current task. The results indicated that the "2-shots" setting achieved the best performance, whereas the introduction of "5-shots" experience led to a decline in performance. This suggested that increasing historical experience did not necessarily enhance the agent's performance. In fact, excessive memory input might occupy a significant portion of the context window, thereby reducing the LLM's capacity to process other relevant perceptual information, causing adverse effects. Additionally, an overabundance of repetitive or low-value memories could confuse the agent, further deteriorating its decision quality.

### 4.4.2 EFFECT OF MULTI-STEP DECIDER

Table 1 analyzed the impact of a multi-step decision-maker based on chain-of-thought (CoT) reasoning on navigation performance. Considering its critical role in the operation of the reflection module, we evaluated the combined effects of CoT and the reflection module under different settings. The results indicated that the presence of CoT and its synergistic interaction with the reflection module had a significant influence on key performance indicators. This was because LLMs often struggled to produce correct answers to complex problems in a single attempt. CoT enhanced their reasoning capabilities by explicitly presenting intermediate reasoning steps before giving a final output, thus

Table 1: Impact of multi-step decider reasoning on navigation performance.

| Settings | | NE↓ | OSR↑ | SR↑ | SPL↑ |
|---|---|---|---|---|---|
| w/o CoT | w/o Reflection | 6.73 | 43.9 | 34.4 | 28.3 |
| | w/ Reflection | 6.61 | 47.9 | 37.4 | 30.0 |
| w/ CoT | w/o Reflection | 5.74 | 53.6 | 46.3 | 36.8 |
| | w/ Reflection | 5.07 | 64.1 | 54.7 | 47.0 |

incrementally approximating the correct solution. These intermediate steps also served as a foundation for reflection, aiding in the identification and correction of errors or deficiencies in the navigation process, thereby improving the agent's navigation performance.

### 4.4.3 EFFECT OF OUTCOME EVALUATOR

Figure 3 illustrated the impact of the outcome evaluator on navigation performance. The data revealed a remarkable improvement in navigation performance when the outcome evaluator was introduced. Specifically, even the most advanced LLMs struggled to accurately identify errors without additional guidance. By incorporating the outcome evaluator, the reflection module could conduct more precise and targeted adjustments and improvements based on past decisions. This elucidated the critical role of the outcome evaluator in enhancing the overall efficacy of the navigation system.

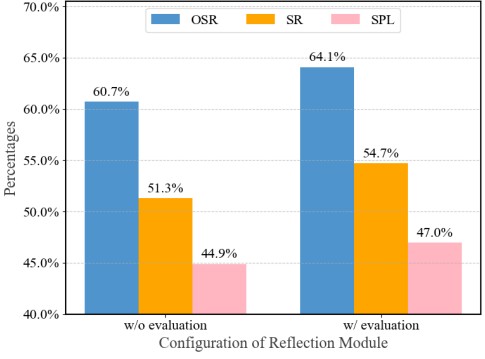

Figure 3: Impact of outcome evaluator on navigation performance.

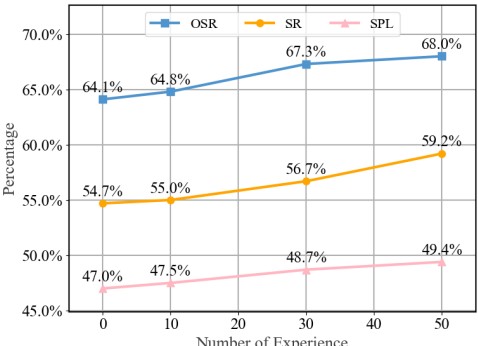

Figure 4: The evaluation of self-evolution ability.

## 4.5 Evaluation of Self-Evolution

As shown in Figure 4, we examined the evolution ability of the VLN framework by adjusting the number of experience entries (set to 0, 10, 30, and 50). As the number of experience increased, the overall performance of SE-VLN showed an improvement trend, i.e., OSR improved from 64.1% to 68.0%, and both SR and SPL were also improved with experience. These indicated that SE-VLN could effectively utilize past experience for continual evolution across different tasks, thereby progressively improving its navigation performance. Moreover, we observed that as the number of tasks increased, the trend of self-evolution gradually flattened out. This might have been due to the limitations of LLMs' reasoning capabilities, and the navigation experience would gradually tend toward homogenization.

## 4.6 Comparison with Existing Approaches

Existing methods were categorized into two types: supervised learning approaches and training-free approaches. As shown in Table 3, we evaluated the navigation performance across various scenarios using a sampled subset of 72 scenes from the R2R dataset. Furthermore, we conducted additional comparisons of the navigation performance between SE-VLN and previous models on a larger validation set that included 11 scenes with 783 trajectories. As shown in Table 2, despite the distributional differences, SE-VLN exhibited slight improvements across key metrics compared to the results from the 72-scene experiments, with further

Table 2: Results on the validation unseen set of the R2R dataset.

| Settings | Methods | NE↓ | OSR↑ | SR↑ | SPL↑ |
|---|---|---|---|---|---|
| Supervised Learning | Seq2Seq Anderson et al. (2018) | 7.81 | 28 | 21 | - |
| | Speaker Follower Fried et al. (2018) | 6.62 | 45 | 35 | - |
| | EnvDrop Tan et al. (2019) | 5.22 | - | 52 | 48 |
| | RecBERT Hong et al. (2021) | 3.93 | 69 | 63 | 57 |
| | DUET Chen et al. (2022) | 3.31 | 81 | 72 | 60 |
| | ScaleVLN Wang et al. (2023) | 2.09 | 88 | 81 | 70 |
| Training-Free | NavGPT Zhou et al. (2024) | 6.46 | 42 | 34 | 29 |
| | TINA Li et al. (2024a) | 5.93 | 48 | 37 | 33 |
| | MapGPT (GPT-4o) Chen et al. (2024a) | 5.65 | 59 | 46 | 34 |
| | SE-VLN (GPT-4o) | 5.03 | 66 | 57 | 50 |

improvements of SR and SPL by 2.3% and 3.0%, respectively. This might be attributed to the fewer number of scenes in the validation set, where the experience repository might contain similar perceptual and decision experience. This result indicated that the experience repository mechanism of the SE-VLN could effectively capture the decision-making logic across tasks. When there were task patterns in new tasks that were similar to those in the experience repository, the agent could quickly transfer and reuse relevant experience, thereby offsetting the performance loss caused by distribution differences.

## 5 Conclusion

In this paper, we proposed a self-evolving visual-language navigation (VLN) framework (SE-VLN) based on multimodal large language models (MLLMs). It achieved self-evolution and eliminated the need for large-scale annotated data training by simulating the advance navigation ability of natural agents. Extensive experiments demonstrated that the SE-VLN achieved state-of-the-art performance through continuous accumulation and reuse of experience, showcasing remarkable self-evolution capabilities. In the future, we plan to explore the introduction of multi-agent collaborative reasoning methods to promote the applications of self-evolving VLN on various fields.

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

# A  APPENDIX

## A.1  THE DETAIL OF EXPERIENCE RETRIEVAL

Specifically, the landmark extractor $\mathcal{F}_{MLLM}$ was first utilized to extract a set of landmark features $\mathcal{L}$ from the current task's instruction $\mathcal{I}$:

$$\mathcal{L} = \{l_j\}_{j=1}^m = \mathcal{F}_{\text{MLLM}}(\mathcal{I}), \quad \text{where } l_j \in \mathcal{V}_{\text{landmark}} \tag{10}$$

Where $\mathcal{V}_{landmark}$ was a predefined landmark vocabulary.

The entity set was then encoded into semantic vectors $\mathbf{q}$:

$$\mathbf{q} = \text{SBERT}\left(\overset{m}{\underset{j=1}{\|}} l_j\right) \in \mathbb{R}^{768} \tag{11}$$

Where $\|$ denoted the sentence concatenation operation, and SBERT$(\cdot)$ represented the Sentence-BERT Reimers (2019) encoder.

Based on this, the cosine similarity between the semantic vector $\mathbf{q}$ and the experience in the experience repository was calculated to retrieve the experience relevant to the current task:

$$\text{sim}(\mathbf{q}, \mathbf{e}_i) = \frac{\mathbf{q} \cdot \mathbf{e}_i}{\|\mathbf{q}\|\|\mathbf{e}_i\|} \tag{12}$$

Where $e_i$ denoted the semantic vector of the $i^{th}$ experience in the experience repository.

Finally, top $N$ experience with the highest similarity scores were selected to construct the "few-shot experience $\mathcal{E}$." These were then injected into the prompt of the reasoning module, enabling the agent to make more informed and reasonable decisions by referencing past experience.

$$\mathcal{E} = \overset{N}{\underset{n=1}{\oplus}} \text{Template}(e^{(n)}) \tag{13}$$

Where $\oplus$ denoted the prompt concatenation operation, and Template$(\cdot)$ converted experience into natural language descriptions.

## A.2  COMPARISON WITH EXISTING APPROACHES

To ensure a fair comparison, we utilized GPT-4o as the inference model for testing both methods. As shown in Table 3, Our results demonstrated that SE-VLN, significantly outperformed MapGPT, the SOTA LLM-powered VLN method. Specifically, the SE-VLN surpassed MapGPT (GPT-4o based) by 0.55 meters in NE, 7.2% in OSR, 8.4% in SR, and 9.2% in SPL. This validated the effectiveness of our approach in improving the navigation performance of LLM-powered agents without the need for training on large-scale datasets.

Table 3: Results on 72 various scenes of the R2R dataset.

| Methods | LLMs | NE↓ | OSR↑ | SR↑ | SPL↑ |
|---------|------|-----|------|-----|------|
| NavGPT Zhou et al. (2024) | GPT-3.5 | 8.02 | 26.4 | 16.7 | 13.0 |
| MapGPT Chen et al. (2024a) | GPT-3.5 | 8.48 | 29.6 | 19.4 | 11.6 |
| MapGPT Chen et al. (2024a) | GPT-4o | 5.62 | 56.9 | 46.3 | 37.8 |
| SE-VLN (ours) | GPT-4o | 5.07 | 64.1 | 54.7 | 47.0 |

## A.3  EVALUATION ON OTHER DATASETS

We further compared the SE-VLN with existing VLN methods on the REVERIE dataset. The test results demonstrated that the existing methods exhibited a significant performance drop on the REVERIE dataset compared to their performance on the R2R dataset, as shown in Table 4. Specifically, we documented the agent's past navigation experience, and the historical data helped the agent avoid repeating past mistakes, thereby enhancing its efficiency and success rate in complex and unknown environments. Through this approach, the SE-VLN not only improved the agent's exploration capabilities in unknown environments but also strengthened its ability to handle challenging tasks, showcasing more intelligent and adaptive behavior.

Table 4: Results on the validation unseen set of the REVERIE dataset.

| Settings | Methods | OSR↑ | SR↑ | SPL↑ |
|---|---|---|---|---|
| Supervised Learning | Seq2Seq Anderson et al. (2018) | 8.1 | 4.2 | 2.8 |
| | RecBERT Hong et al. (2021) | 23.1 | 30.7 | 24.9 |
| | DUET Chen et al. (2022) | 50.0 | 45.8 | 35.3 |
| Training-Free | NavGPT Zhou et al. (2024) | 28.3 | 19.2 | 14.6 |
| | MapGPT (GPT-4o) Chen et al. (2024a) | 36.9 | 30.6 | 22.3 |
| | SE-VLN (GPT-4o) | 43.7 | 35.2 | 24.8 |

## A.4 QUALITATIVE ANALYSIS

Figure 9 and 10 illustrated examples of the prompts required for SE-VLN's decision per frame and case studies of leveraging navigational experience to aid decision, respectively. As shown in the figure, at each step, SE-VLN updated its verbal topological map in real-time and formulated prompts. By comprehensively analyzing the input information, it generated a series of thinking processes and planning strategies, leading to the most suitable executing decisions. For instance, in step 2, when two vantage points could both lead to the destination, SE-VLN reviewed past navigational experience in similar tasks to identify which path might result in fewer obstacles or a more optimal navigation route. Such a strategy enabled SE-VLN's navigational performance to continuously improve as experience accumulated.

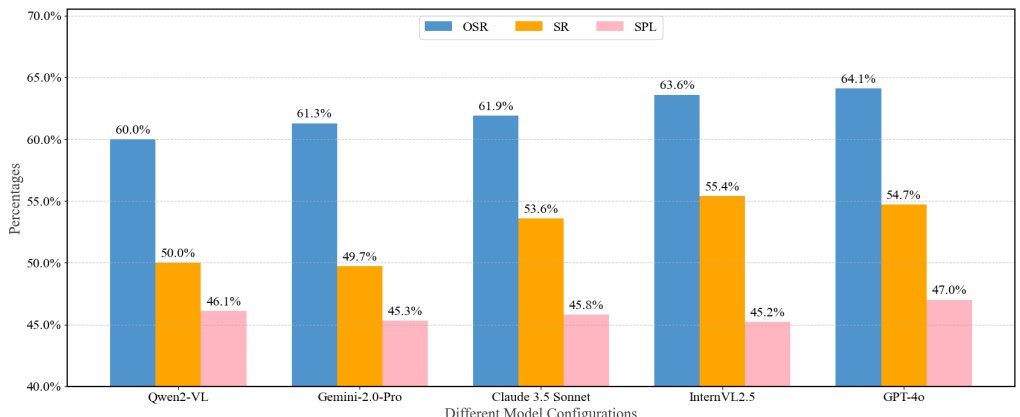

Figure 5: Performance of Different MLLMs on the R2R Sampling Subsets.

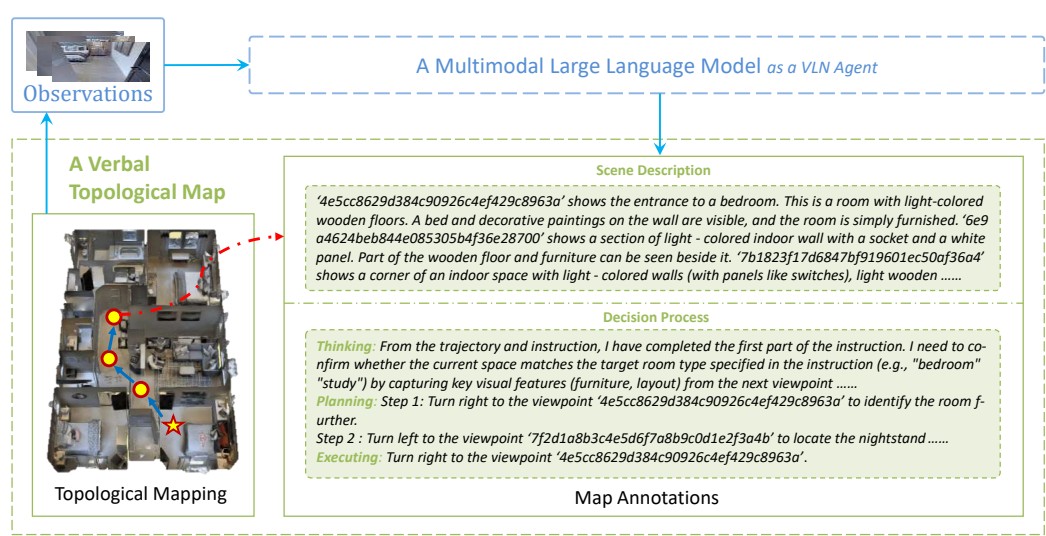

Figure 6: The pipeline of the verbal topological map.

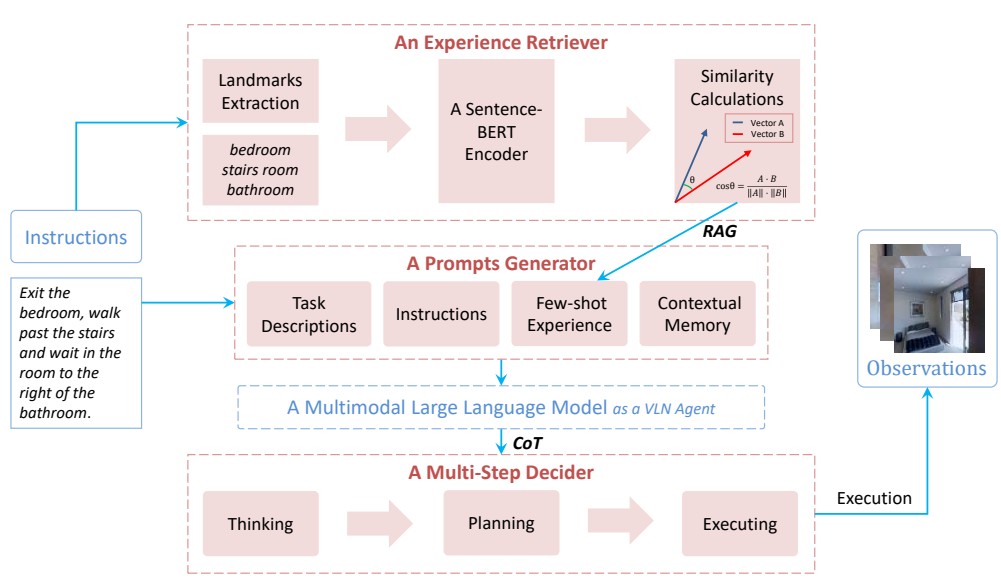

Figure 7: The pipeline of the retrieval-augmented thought-based reasoning module.

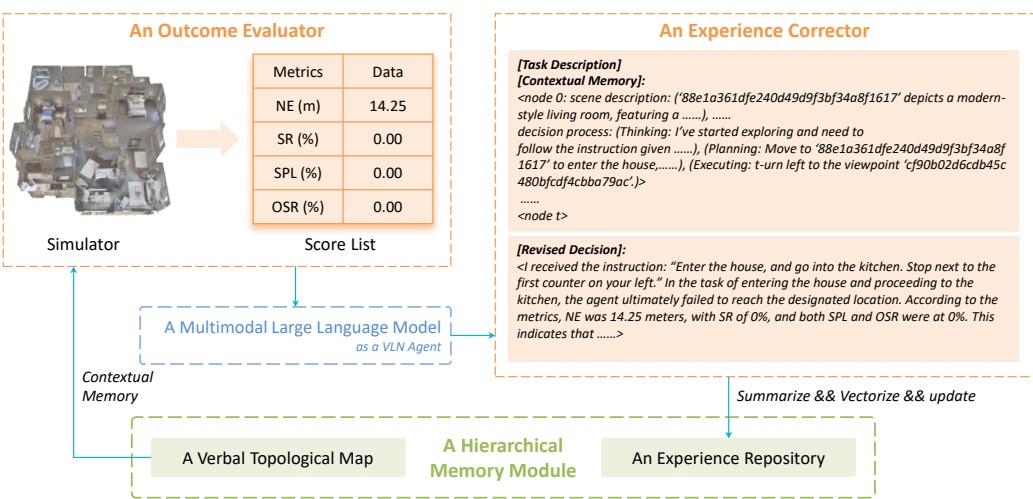

Figure 8: The reflection module workflow. The agent computed the navigation performance of current task, i.e., score list, through an outcome evaluator and integrated this with the contextual memory of the verbal topological map for targeted reflection. Subsequently, the revised decision was updated into the experience repository as experience.

```
$Prefix:
Task Description:
[Task background]
As an embodied robot, you need to explore and navigate the environment according to human language instructions until you reach the
destination. At each step, you will receive visual observations of the environment, and you need to parse this visual observation data to
determine the best navigational viewpoint to achieve navigation.
[Input Definitions]
"Instruction" refers to a global guidance that you need to follow step by step to execute navigation according to the requirements of the
instruction.
"Landmark Features" refers to specific room types, such as bedrooms, living rooms, and kitchens.
"Scene Description" refers to the linguistic description of the current location. You need to convert each step's visual input into a
text description.
"Decision Process" refers to how you plan step by step through thinking during navigation and generate predictions of actions. It
consists of "thought", "planning" and "action" three parts.
"Revised Decision" refers to the decision to adjust or improve upon the identification of errors or non-optimal choices during previous
navigation.
"Few-shot Experience" refers to memories of past navigation, containing the decision-making processes of similar tasks in the past. You
can refer to it for decision-making. It includes four parts: landmark features, scene description, decision process and revised decision.
It can be expressed in the following forms:
< landmark features >
< step 0: "scene description" , "decision process" >
< …… >
< step t: "scene description" , "decision process" >
< revised decision >
"Verbal Topological Map" refers to a textual representation method that records the navigation process. It includes the linguistic
descriptions of your visual observations at each position and the multi-step decision-making process. The decision process for each node
is updated before moving to the next node. It can be expressed in the following forms:
< node 0: "scene description", "decision process" >
< …… >
< node t: "scene description", "decision process" >
[Output Requirements]
For each step of the decision, you should combine "instruction", "scene description", "few-shot experience", and "verbal topological map"
to make multi-step decisions, and choose the most appropriate viewpoint to navigate. "Multi-step decision" means that you need to think
step by step to predict the final executed action. It includes three steps: "thinking", "planning" and "executing". Therefore, your
answer must include these three steps. After clarifying the target location, you need to try to move to within 1 meter of the destination
and stop. The output action needs to include action and destination, i.e., viewpoint, such as "Go straight to the viewpoint
'e1166572e82f43d5857232295dcf4879'.".

$Input:
Instruction: Head straight until you pass the wall with holes in it the turn left and wait by the glass table with the white chairs.
Few-shot Experiences:
< Shot 1 >, < Shot 2 >, ……, < Shot k >
Verbal Topological Map:
< node 0: "scene description", "decision process" >
……
< node t: scene description: ('7cb2c4bc9ad84463b4d0c6a91b8c4640': The viewpoint shows a metal screen and an open doorway leading to
another room.),…… >
```

Figure 9: The prompts for the retrieval-augmented thought-based reasoning module consisted of two parts: the prefix prompt and the input prompt. These respectively provided the prerequisite information needed for the task and specified the input required at each step.

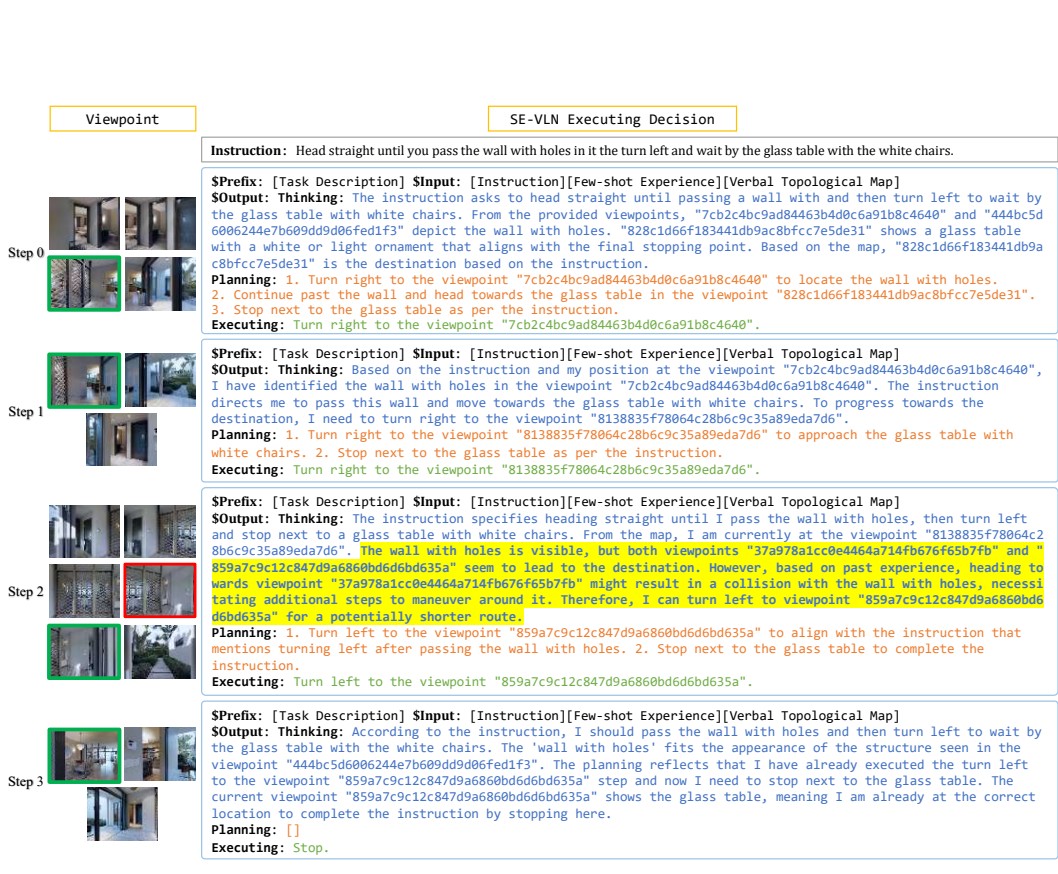

Figure 10: A successful case of SE-VLN which by referencing past experience, effectively avoided potential collisions (red box) and optimized path selection, ultimately achieving navigation via the shortest path (green box).

