# OpenReview forum: "SE-VLN: A Self-Evolving Vision-Language Navigation Framework Based on Multimodal Large Language Models"
_ICLR.cc/2026/Conference — ICLR 2026 Conference Withdrawn Submission_

### Official Review · Reviewer_ypdp · 2025-10-16

**Soundness:** 2
**Presentation:** 3
**Contribution:** 2
**Rating:** 4
**Confidence:** 4

**Summary:**

This paper proposes a self-evolving vision-language navigation framework, SE-VLN, that enables VLN agents to continuously evolve during testing without additional training. The framework consists of three core modules: (1) a hierarchical memory module that stores short-term visual observations and decision-making processes via a verbal topological map, and long-term experience in an experience repository; (2) a retrieval-augmented thought-based reasoning module that retrieves relevant historical experience from the repository and combines chain-of-thought prompting for multi-step decision-making; and (3) a reflection module that analyzes agent decisions based on task evaluation results to enable continuous evolution. SE-VLN achieves navigation success rates of 57% and 35.2% on R2R and REVERIE datasets in unseen environments, representing relative performance improvements of 23.9% and 15.0% over state-of-the-art methods, respectively.

**Strengths:**

- **Quality**: The experimental evaluation is thorough, with comprehensive results on two standard VLN benchmarks (R2R and REVERIE). The performance improvements (23.9% and 15.0% relative gains) are statistically significant and well-documented. The framework's ability to improve with increasing experience repository size provides strong evidence of its self-evolving nature.
- **Clarity**: The paper is well-structured and clearly written. The three core modules are logically explained, and the workflow is effectively illustrated in Figure 1. The hierarchical memory module, retrieval-augmented thought-based reasoning, and reflection module are conceptually well-defined and consistently applied throughout the paper.
- **Significance**: The work addresses a critical limitation in current VLN systems—lack of continuous evolution capability. The proposed framework offers a promising direction for building more adaptable and efficient VLN agents that can learn from experience without additional training.

**Weaknesses:**

- **Ambiguity around the “training-free” claim**: The paper prominently claims SE-VLN is “training-free”, but this assertion lacks sufficient clarification. Specifically, it is unclear how the experience repository and other modules are initialized. If the repository is constructed from prior navigation episodes on the R2R or REVERIE datasets (the same benchmarks used for evaluation), then the reported performance gains may partly reflect data leakage or overfitting to those environments, rather than true generalization to unseen settings.
- **Ambiguity in the modules' implementation**: While the three core modules (hierarchical memory, retrieval-augmented reasoning, and reflection) are described as operating without parameter updates, it remains ambiguous whether any component relies on pretrained or fine-tuned models that were trained on in-domain VLN data. And are they using separate models or integrated within the MLLM?
- **Limited computational analysis**: The paper demonstrates performance improvements but does not discuss the computational cost of the SE-VLN framework compared to baseline zero-shot methods. How much additional computational resources are required for the self-evolving capability? Could the authors provide a detailed analysis of inference time and resource requirements?
- **Lack of detailed comparison with the most recent VLN methods**: While the paper compares with some methods, most of the them are outdated. It would be beneficial to include a more detailed comparison with the latest VLN approaches to better establish the novelty and relative contribution of SE-VLN.
- **Novelty**: The paper claims to be “the first training-free self-evolving VLN framework”. While some recent concurrent efforts such as EvolveNav and CorrectNav also explore experience-based self-correction or iterative improvement in VLN. What are the essential differences between SE-VLN and these approaches? Moreover, the framework appears to combine established techniques (RAG, chain-of-thought, episodic memory) in a modular pipeline, without demonstrating that their integration itself yields emergent capabilities beyond the sum of parts. A more in-depth ablation and a dedicated discussion about SE-VLN’s design choices would be crucial to clarify the contribution and justify the novelty claim.

**Questions:**

See Weaknesses

---

### Official Review · Reviewer_3gAQ · 2025-10-28

**Soundness:** 2
**Presentation:** 2
**Contribution:** 2
**Rating:** 4
**Confidence:** 4

**Summary:**

This paper proposes a SE-VLN framework based on a MLLM. The core contribution is a framework that enables a VLN agent to continuously learn and evolve from both successful and failed experiences during testing without requiring additional training, achieved through a hierarchical memory module, a retrieval-augmented reasoning module, and a reflection module. The method demonstrates superior performance over existing approaches on R2R and REVERIE datasets and shows a continuous performance improvement capability as experience accumulates. However, the justification for its novelty and some technical details need strengthening.

**Strengths:**

1. As a training-free method, it leverages the capabilities of a pre-trained MLLM, avoiding the need for large-scale annotated data and the complexity of model fine-tuning iterations.

2. The experimental section is comprehensive, covering mainstream datasets (R2R, REVERIE), multiple MLLM base models, and a crucial evaluation of the self-evolution capability.

**Weaknesses:**

1. While the concept of "self-evolution" is appealing, its implementation mechanism  "Reflection + Memory Update" is not entirely novel within the field of LLM-based Agents. Numerous existing works have explored continuous learning and policy optimization for LLM agents through reflection and memory mechanisms. The paper needs to more clearly delineate its specific innovations compared to this established body of work.

2. The RAG strategy appears oversimplified. The Experience Retriever relies solely on ​​core landmark features​​ from the instruction for similarity matching. In complex indoor environments, relying only on landmarks (e.g., "bedroom", "kitchen") can easily lead to retrieving experiences that are too generic or irrelevant, potentially limiting the effectiveness of the retrieved context.

3. Figure 2 indicates that "2-shots" performance is best, while "5-shots" performance declines. The authors attribute this to context window limitations and redundant/low-value memories. This is a critical observation, but the current analysis remains superficial. More in-depth ablation experiments and analysis are necessary to substantiate this claim and explore the underlying reasons.

4. Although a project page link is provided, the code and models are currently inaccessible, which hinders verification and reproducibility.

**Questions:**

Please refer the above Weaknesses.

**Details Of Ethics Concerns:**

None.

---

### Official Review · Reviewer_FChe · 2025-10-29

**Soundness:** 2
**Presentation:** 3
**Contribution:** 2
**Rating:** 4
**Confidence:** 5

**Summary:**

This paper proposes **SE-VLN**, a self-evolving framework for vision-language navigation (VLN) powered by multimodal large language models (LLMs). SE-VLN innovatively integrates a hierarchical memory module, a retrieval-augmented reasoning module, and a reflection module to enable continual evolution during testing.  **SE-VLN**  achieves significant performance improvements on R2R and REVERIE 、benchmarks, showing adaptability and scalability as the experience repository grows.

**Strengths:**

1. **Well-Presented and Clear**: The paper is written in a clear and accessible manner, ensuring that readers can easily understand, reproduce, and validate the proposed methods.
2. **Performance Improvements**: The framework achieves significant performance improvements on offline VLN environments.

**Weaknesses:**

1. **Incremental Contribution**: The proposed framework combines existing techniques like retrieval-augmented generation (RAG), chain-of-thought (CoT), and reflection mechanisms to tackle VLN tasks on offline datasets. While it achieves performance improvements, the approach feels incremental and lacks distinctive insights or novel designs, falling short of the quality standards typically expected at conferences like ICLR.

2. **Unfair Comparisons**: Since zero-shot methods in this paper rely on API calls to proprietary models, which are constantly updated and improved, the comparison with earlier methods using older versions of these models is inherently unfair. As a result, the conclusions drawn may not fully reflect the significance of the contributions.

3. **Toy-Level Problem**: VLN tasks require high practical feasibility, especially for real-world applications. This study limits itself to offline simulation experiments on a small amount of trajectory data, which fails to demonstrate the robustness of the algorithm. To truly highlight its value, the framework should be trianed with agentic reinforcement learning approaches or validated in real-world physical environments, which would significantly enhance its utility and contribution.

**Questions:**

**Concern on Claim of Self-Evolution**

Self-evolution typically refers to a model's ability to continually learn and enhance itself through iterative cycles of data generation and model improvement in an interactive environment. However, the framework proposed in this paper relies on non-trainable, API-based models without iterative feedback loops between data and model enhancement. This setup does not fulfill the essential conditions of self-evolution, as there is no mutual reinforcement between the data and the model.

Thus, the claim of self-evolution in the paper may appear inappropriate or misleading. A more accurate description of the approach would be as a static offline adaptation framework, rather than a genuinely self-evolving architecture. Clear differentiation from true self-evolution methods is necessary to avoid confusion.

---

### Official Review · Reviewer_MhBr · 2025-10-30

**Soundness:** 3
**Presentation:** 2
**Contribution:** 3
**Rating:** 2
**Confidence:** 4

**Summary:**

This paper introduces SE-VLN, a training-free framework for Vision-Language Navigation (VLN) designed to mimic the continuous learning and adaptation seen in natural agents. It aims to solve the problem of existing VLN methods, which rely on the fixed knowledge and static reasoning abilities of Large Language Models (LLMs) and lack an efficient evolutionary capacity.
To this end, SE-VLA is built based on three core modules: a hierarchical memory module, a retrieval augmented thought-based reasoning module, and a reflection module. Experiments on the R2R and REVERIE datasets show that SE-VLN achieves state-of-the-art success rates (57% on R2R, 35.2% on REVERIE) among training-free methods. The results also demonstrate that the agent's performance improves as the Experience Repository accumulates more entries, confirming its self-evolving capability.

**Strengths:**

1. The framework's primary strength is its closed-loop, self-evolving process. Unlike methods that rely on static datasets, SE-VLN actively learns from its mistakes.
2. The organization of this paper is logical.

**Weaknesses:**

1. The paper's core motivation is to emulate the "autonomous evolution" of "natural agents" (e.g., horses, migratory birds) to overcome "data dependency". However, the proposed method's "Reflection Module" is entirely dependent on an omniscient oracle. The "Outcome Evaluator" requires "ground truth data from the MatterPort3D simulator", i.e., $\tau_{gt}$ to calculate metrics and identify failure. This is a form of strong supervision, not the autonomous, experience-driven adaptation of a "natural agent." This contradicts the concept of self-evolution.
2. One of my main concerns is that SE-VLN is more like a composite of well-established technologies. The "Retrieval-Augmented Thought-Based Reasoning Module" explicitly combines Retrieval-Augmented Generation (RAG) and Chain-of-Thought (CoT). Similarly, the "Reflection Module" is a concept explored in prior agent-learning literature. This makes the contribution feel more like a sophisticated engineering and integration effort rather than a fundamental methodological breakthrough. The authors are expected to provide more discussion.

**Questions:**

Please refer to the Weaknesses.

---

### Official Review · Reviewer_n3tp · 2025-11-01

**Soundness:** 3
**Presentation:** 2
**Contribution:** 2
**Rating:** 4
**Confidence:** 3

**Summary:**

This paper proposes SE-VLN, a training-free, self-evolving vision-language navigation (VLN) framework that leverages multimodal large language models (MLLMs) to enable continual improvement during test time without retraining. It addresses the limitations of existing VLN approaches, which rely on static knowledge and lack mechanisms to accumulate and reuse experiential knowledge for improved scene generalization. The main contributions include (i) a hierarchical memory module with a verbal topological map and long-term experience repository, (ii) a retrieval-augmented, thought-based multi-step reasoning module combining RAG and chain-of-thought planning, and (iii) a reflection module for error analysis and experience correction. LLM baselines are considered, with comparisons against multiple MLLMs (e.g., GPT-4o, Claude, Gemini) and prior VLN methods.

**Strengths:**

1. Introduces a training-free, self-evolving VLN framework that continuously improves during deployment, a relatively unexplored capability in LLM-powered agents.
2. Well-structured architecture combining hierarchical memory, retrieval-augmented reasoning, and reflection aligns closely with human-like learning and adaptability.
3. Comprehensive experimental evaluation on multiple datasets (R2R, REVERIE) and with diverse MLLMs, demonstrating gains over prior methods.

**Weaknesses:**

1. The framework is complex, with multiple handcrafted components, which may reduce accessibility and reproducibility for the broader community.
2. Heavy reliance on specific LLM capabilities (e.g., GPT-4o) risks reduced generality if applied to smaller or weaker models.
3. Limited discussion on computational cost, scalability, and resource requirements for maintaining large experience repositories during test-time deployment.
4. Evaluation is confined to indoor Matterport3D environments; out-of-domain generalization to other embodied tasks remains untested.

**Questions:**

1. How sensitive is SE-VLN’s performance to the choice of underlying MLLM?
2. What are the computational and latency trade-offs of running retrieval, prompt generation, and reflection in real-time navigation scenarios?
3. How would the proposed framework perform in more dynamic or outdoor VLN environments not well-represented by Matterport3D?
4. Potential missing reference: Experiential Co-Learning of Software-Developing Agents (ACL2024)

---

### Note · Authors · 2025-11-26

I have read and agree with the venue's withdrawal policy on behalf of myself and my co-authors.